# Cellular Changes in Injured Rat Spinal Cord Following Electrical Brainstem Stimulation

**DOI:** 10.3390/brainsci9060124

**Published:** 2019-05-28

**Authors:** Walter J. Jermakowicz, Stephanie S. Sloley, Lia Dan, Alberto Vitores, Melissa M. Carballosa-Gautam, Ian D. Hentall

**Affiliations:** 1Department of Neurological Surgery, University of Miami, 1095 NW 14th Terr, Miami, FL 33136, USA; 2Miami Project to Cure Paralysis, University of Miami, 1095 NW 14th Terr., Miami, FL 33136, USA; ssloley@gmail.com (S.S.S.); blurryblue@gmail.com (L.D.); alb9402@hotmail.com (A.V.); mcarballosa@gmail.com (M.M.C.-G.); IHentall@med.miami.edu (I.D.H.)

**Keywords:** neuromodulation, inflammation, serotonin, neural progenitor cell, deep brain stimulation

## Abstract

Spinal cord injury (SCI) is a major cause of disability and pain, but little progress has been made in its clinical management. Low-frequency electrical stimulation (LFS) of various anti-nociceptive targets improves outcomes after SCI, including motor recovery and mechanical allodynia. However, the mechanisms of these beneficial effects are incompletely delineated and probably multiple. Our aim was to explore near-term effects of LFS in the hindbrain’s nucleus raphe magnus (NRM) on cellular proliferation in a rat SCI model. Starting 24 h after incomplete contusional SCI at C5, intermittent LFS at 8 Hz was delivered wirelessly to NRM. Controls were given inactive stimulators. At 48 h, 5-bromodeoxyuridine (BrdU) was administered and, at 72 h, spinal cords were extracted and immunostained for various immune and neuroglial progenitor markers and BrdU at the level of the lesion and proximally and distally. LFS altered cell marker counts predominantly at the dorsal injury site. BrdU cell counts were decreased. Individually and in combination with BrdU, there were reductions in CD68 (monocytes) and Sox2 (immature neural precursors) and increases in Blbp (radial glia) expression. CD68-positive cells showed increased co-staining with iNOS. No differences in the expression of GFAP (glia) and NG2 (oligodendrocytes) or in GFAP cell morphology were found. In conclusion, our work shows that LFS of NRM in subacute SCI influences the proliferation of cell types implicated in inflammation and repair, thus providing mechanistic insight into deep brain stimulation as a neuromodulatory treatment for this devastating pathology.

## 1. Introduction

Spinal cord injury (SCI) involves widespread damage of local and distal neuronal networks [1,2]. It is a major cause of disability worldwide, leading to impairments in motor and autonomic function as well as debilitating mechanical allodynia [3,4,5]. A key difficulty in treating SCI arises from the diversity of cellular processes and physiological functions affected. However, since some degree of functional improvement occurs naturally in the weeks following injury, it is possible that the sequelae of SCI may be overcome to some extent by interventions that enhance endogenous beneficial processes [2,6,7]. Chronic low-frequency electrical stimulation (LFS) of various brain and spinal targets has long been known to influence the perception of chronic pain in select patients [8,9,10]. However, more recently, LFS of several of these targets and their inputs has received attention for its ability to improve functional outcomes in laboratory and clinical models of SCI [6,11,12,13,14,15]. A critical impediment to the clinical adoption of these treatments is lack of insight into the therapeutic mechanisms involved.

We have previously shown that LFS of the nucleus raphe magnus (NRM) or its primary midbrain afferent, the periaqueductal gray (PAG), improves functional and anatomic recovery from incomplete SCI in rats, while increasing myelination and expression of serotonin- and CGRP-containing axons around the injury site [6,16]. This includes improvements in cutaneous mechanical allodynia [15]. Other sequelae of SCI, such as reduced gastric motility and autonomic dysreflexia, appear to be influenced with LFS of NRM as well [16]. This has led to the proposal that the raphe nuclei of the brainstem, which collectively constitute a system of fibers that diffusely project to most areas of the nervous system, form a key link in a centralized restorative feedback system. The NRM provides a major spinal projection for this system. In support of the generality of this proposed repair model, LFS in the midbrain raphe has been found to improve outcomes in a rat model of traumatic brain injury while LFS in the mouse’s NRM ameliorates experimental autoimmune encephalitis (EAE) [14,17].

The mechanisms invoked by brainstem LFS are likely multiple, addressing the multifarious nature of traumatic injuries [2,7,18]. We have previously shown that 2 h of NRM LFS three days after incomplete SCI restores cyclic adenosine monophosphate (cAMP), an intracellular signaling molecule implicated in inflammation and repair, to pre-injury levels [19]. This effect on cAMP is blocked by the non-specific 5-HT_7A_ antagonist pimozide and is accompanied by increases in phosphorylated PKA and CREB, as well as changes in the expression of various genes implicated in inflammation and repair [6,17,19]. We hypothesize that the result is an activation of conserved pathways that blunt secondary injury and enhance restorative processes. To further test this concept, the present study explores whether LFS of NRM initiates changes in the proliferation of immune and neuroglial progenitor cells after incomplete bilateral SCI in the rat. 

## 2. Materials and Methods

### 2.1. Animal Procedures

This study was conducted under protocols approved by the local Institutional Animal Care and Use Committee. Subjects were young adult female Sprague–Dawley rats (225–250 g), all with injuries. Animals were randomly assigned to two groups: Control (*n* = 11) and LFS (*n* = 10). One rat in the LFS group was lost unexpectedly 12 h after injury. Subgroups were randomly selected for certain markers: eight control and six LFS for the NG2 stain and six control and five LFS for the Blbp/Sox2 stains. 

Surgical procedures are described in detail elsewhere [17]. Briefly, under isoflurane anesthesia rats were placed in a stereotaxic holder and cervical laminectomy performed. An Infinite Horizon Impactor (Lexington, KY) produced a midline C5 contusion with a force of 200 kdyn, velocity of 1 m/s, tissue displacement of 0.5 mm, and contact diameter of 2.5 mm. Following impact, the overlying muscle and skin were approximated with nylon sutures. Animals then received a stimulator, which was not activated until 24 h after injury (Figure 1A,B). For controls, stimulators were implanted but never activated. 48 h after injury animals were given a single intraperitoneal dose of bromodeoxyuridine (BrdU, 50–70 μg). Seventy-two hours after injury rats underwent phenobarbital euthanasia and paraformaldehyde perfusion.

### 2.2. Stimulators

To place stimulators [20], a craniostomy was drilled dorsal to the NRM (2.2 mm caudal, 0 mm lateral, and 1 mm ventral to the interaural line in the horizontal plane). The wireless epoxy-encapsulated stimulator, which measures 12 × 8 × 5 mm and weighs 2 g, was fixed to the skull with screws and dental cement. Two electrodes protrude directly from the capsule: A tungsten microelectrode (cathode) with a resistance of 0.5 megohm at 1 kHz targeted to the NRM, and a stainless-steel wire (anode) attached to a skull screw. Parameters are controlled by a pulsed magnetic field, and it communicates its status via width-modulated infrared pulses received by a custom detector. Status was checked twice daily during the two days animals received LFS. The output consisted of fixed-amplitude 30 µA monopolar pulses with a width of 1.0 ms. The microprocessor cycled between 5 min of 8 Hz stimulation and 5 min of rest during 12 daylight hours [6,14,16,17]. 

### 2.3. Histological Analysis

Spinal cords were extracted, post-fixed overnight, embedded in paraffin, and sectioned coronally at 12 μm. Slides were deparaffinized with xylene and rehydrated with ethanol and phosphate buffered saline (pH 7.4). Hematoxylin and eosin staining was used to grossly visualize spinal cord lesions (Figure 1C). For immunohistochemical analyses, antigen retrieval was performed in a steam bath with sodium citrate buffer (pH 6.0) for 20 min. Immunostaining was carried out for BrdU, Sox2, Blbp, NG2, GFAP, CD68, iNOS, arginase-1 (arg-1), doublecortin (DCX), NeuN, and APC (Figure 2, Appendix A for antibody details). To reveal BrdU, slides were placed in 4% paraformaldehyde for 20 min, followed by 2 M HCl solution at 37 °C for 30 min prior to application of the primary antibody. Primary antibodies were applied overnight, and secondary antibodies were applied at 1:500 dilution for 2 h. The following markers were co-labeled with BrdU: CD68, GFAP, NG2, Sox2, and Blbp. Other co-labeling combinations were CD68 with Arg1 and iNOS, and NG2 with APC. Staining of nuclei with DAPI was sometimes included. Negative controls using isotype-matched antibodies were examined for each batch of slides.

Cell counts were done by an investigator blinded to animal treatments using epifluorescence microscope-based stereology (Stereo Investigator, Williston, VT). At the level of the lesion, three sections spaced 21 µm apart were used for stereology. For analyses of BrdU, CD68, and GFAP in caudal or rostral tissue, one section 0.75 cm from the lesion epicenter was examined for each location. Separate regional counts were obtained for dorsal white (DWM), ventral white (VWM), and grey matter (GM) after manually tracing these regions at 10×. For each of the three sections, a sampling grid of 225 × 225 μm was placed over each of the three contoured subregions. Within each grid box an area of 75 × 75 μm was selected by the software as the counting frame and cell markers were counted at 60×. The number of counting frames used for each subregion varied between sections but typically ranged from 60 to 90 for DWM, 120 to 180 for GM, and 190 to 270 for VWM. 

To examine whether injuries delivered to LFS and control animals differed in intensity, volumetric analyses of lesion cavities was done by tracing sections spaced by 105 µm and aggregating them into a single 3D structure using Callesion software (v1.03) [21]. 

### 2.4. Statistical Analyses

Quantitative analyses were performed using SPSS (IBM Corporation, Armonk, NY, USA) or spreadsheet calculations, with *p* ≤ 0.05 used as the criterion for significance. Comparisons of means, whether for two or three groups, were performed using bootstrap analysis, by resampling data with replacement 1000 times. Multivariate analyses of treatment effects were performed using multinomial logistic regression. In order to have a complete dataset for multivariate analyses, we included only the six LFS and five control rats that were stained for Blbp/Sox2. Stepwise logistic regression analysis was used to define multivariate models that best captured LFS effects. 

## 3. Results

### 3.1. Did SCI Lesions Differ Between LFS and Control Groups?

Following moderate C5 contusion, flaccid, incomplete quadriplegia was always observed. All animals had shown some recovery of function by the time the animals were euthanatized. Histological analysis showed small dorsally located cavities with larger surrounding zones of necrotic tissue. Lesion cavities were generally confined to the DWM and GM, with minimal extension observed into VWM. Volumetric analysis revealed no difference in volume of the lesion cavity between LFS and control animals (0.075 ± 0.016 mm^3^ vs. 0.077 ± 0.016 mm^3^, *p* = 0.38). 

### 3.2. Were Cellular Effects of LFS Specific to the Injury Site? 

BrdU, CD68, and GFAP cell counts were all highest at the lesion as opposed to rostrally or caudally (Table 1, above). The association of LFS with changes in cell counts tended to be greatest at the lesion (*p* = 0.10, three-factor bootstrap). With all spinal cord subregions included, pairwise bootstrap analysis revealed significantly fewer CD68-positive cells (*p* = 0.05), but not BrdU-positive cells (*p* = 0.08), at the lesion in LFS vs. control animals. Significant differences in the counts of these cell markers were not observed for LFS vs. control animals rostral or caudal to the lesion. 

When cell counts were compared for the different spinal cord subregions at the level of the lesion, their association with LFS was consistently greater at DWM and GM than at VWM (Table 1, below). Given this finding that LFS cellular effects localized predominantly to DWM and GM and the observation that the lesions were confined to DWM and GM histologically, these two subregions were grouped for subsequent analyses (see Appendix A for analyses performed with all spinal cord subregions included).

### 3.3. Was Immune and Neural Progenitor Cell Expression at the Injury Site Influenced by LFS?

In univariate analysis of the relationship between NRM LFS and the expression of immune and neuroglial progenitor markers, CD68-positive cells were significantly lower in LFS vs. control animals (Figure 3A). CD68-positive cells co-labeled with iNOS were higher in the LFS animals, but CD68 cells co-labeled with arginase-1 did not significantly differ (*p* = 0.20) (Figure 3B,C). GFAP-positive cells typically showed an astrocytic morphology with one or more long processes and with no significant differences in the number of processes between LFS and control animals at the lesion (Figure 3D). In the caudal but not the rostral or lesioned region, the number of GFAP positive cells was increased in the LFS group. NG2-staining cells at the lesion, which typically co-expressed APC suggestive of an oligodendrocyte fate, did not differ between the control and LFS animals (Figure 3E) [7]. Sox2 was predominantly observed in nuclei of ependymal and subependymal cells of the central canal, though positive cells were distributed throughout most areas of the injured cord. The expression of Sox2 was reduced in the LFS animals by about 50%, while Blbp conversely was roughly doubled (Figure 3F,G). Doublecortin-positive cells were not seen in the injured spinal cord.

Multivariate logistic regression analysis using all immune and neuroglial progenitor cell markers implied a significant LFS-related effect (F_7_ = 20.0, *p* = 0.016), and stepwise logistic regression gave the following best-fit model for the relation between NRM LFS and changes in cell counts after SCI (F_3_ = 69.2, *p* < 0.001): NRM LFS ≈ Sox2 + Blbp + CD68(1)

### 3.4. Was LFS Associated with Changes in Cellular Proliferation?

The total BrdU count was lower at the lesion for rats treated with LFS of NRM compared to the control animals (Figure 4A–C). At the lesion, BrdU co-labeled with many of the markers examined, including CD68, GFAP, NG2, Blbp, and Sox2, although NeuN did not co-localize with BrdU (Figure 4D–H). CD68 and BrdU co-labeled cells did not achieve statistical significance (Figure 4D) (*p* = 0.07). Univariate analysis indicated a significant relation between LFS and changes in the expression of cells co-localized with Sox2 and BrdU, which were lower in the LFS animals (Figure 4G). Multivariate logistic regression analysis with all regressors found a significant relationship between LFS and the expression of all BrdU co-labeled cells (F_6_ = 7.8, *p* = 0.033), prominently involving CD68/BrdU (*p* = 0.055) and Blbp/BrdU (*p* = 0.056), although stepwise logistic regression did not produce a model with a significant fit for the effect of LFS on BrdU co-labeled cells. 

## 4. Discussion

The present study suggests that treatment of incomplete cervical SCI with two days of LFS in the NRM leads to changes in cellular proliferation in the subacute injury period. Contrary to our expectations, post-injury proliferation was reduced, which we attribute to the treatment reducing inflammation. Other published studies have also shown immune effects of direct or indirect brainstem neuromodulation. LFS of brainstem (PAG, NRM, C1 neurons) or its inputs (vagal nerve, fastigial nucleus) is anti-inflammatory in diverse models of neurologic (Stroke, SCI) and non-neurologic (rheumatoid arthritis, Crohn’s disease, sepsis) illness [17,22,23,24]. Studies of analgesic LFS have also implicated anti-inflammatory mechanisms [25]. In a murine EAE model, our laboratory found that stimulation of NRM prevents disease exacerbation and is associated with reduced spinal cord immune infiltration and cytokine expression [17]. Similarly, vagal nerve stimulation has been shown to reduce disease burden in clinical trials for multiple sclerosis. This neuroinflammatory reflex has been proposed to involve the autonomic nervous system and hippocampal-pituitary-adrenal axis [22,23,24], but the serotonergic system may be involved as well, as it has been linked to immune control. Serotonin and its receptors are present on numerous cells of the adaptive and innate immune systems and are implicated in microglial activation [26,27,28].

Although the decreased expression of CD68 is suggestive of anti-inflammatory stimulation effects, the increased co-expression of iNOS and CD68 is inconsistent with this, since iNOS is traditionally a marker of pro-inflammatory macrophages [2,29]. This is likely due to the complex dynamics of macrophage polarization in vivo and our use of the three-day time point. Recent work suggests that sequential activation of both M1 (pro-inflammatory) macrophages, which predominate the first few days after injury, and M2 (anti-inflammatory) macrophages is important for repair after injury [30,31]. 

Our work also suggests that NRM stimulation influences neuroglial progenitors after SCI. Radial glia, which are increased by LFS in this study, serve as scaffolds for neuronal migration during development and can give rise to new neurons [18,29,30]. Radial glia are associated with repair after injury and they improve functional recovery when injected after SCI in rats [18]. The decrease in Sox2 was unexpected. Sox2 is expressed in immature NPCs of a neuronal lineage. It is co-expressed with Blbp, though Blbp expression is lost earlier in neuronal development. Sox2 maintains the proliferative and developmental potential of NPCs and inhibits the progression from radial glial cell to immature neuron [18,30,31]. We speculate that the depletion of Sox2-positive NPCs here could be caused by increased progression towards a neuronal fate without adequate short-term replenishment, or by the shifting of regenerative resources away from the production of neurons. 

The present findings as a whole provide additional support to our hypothesis that the brainstem raphe nuclei are key central links in restorative feedback responses to injury, potentially playing a role in the restorative and analgesic effects of LFS reported previously [6,14,16,17]. Consistent with this model, raphe neurons respond to physiologic correlates of injury, such as pain, low blood pressure, and circulating cytokines, and send diffuse projections to all regions of the neuraxis [26,32]. Although our stimulation protocol precludes elucidation of the precise circuitry, our prior work shows that LFS of NRM leads to synaptic release of serotonin at the SCI site and that therapeutic effects are blocked with a 5-HT_7A_ antagonist, suggesting serotonin may be involved [19]. Serotonin is crucial to the regulation of cell survival during development and is implicated in various protective and trophic processes [26,33,34]. Raphe neurons also secrete several neuropeptides, including galanin, substance P, and thyrotropin-release hormone, that are potentially protective after SCI [32,33]. The relation of NRM LFS to other stimulation targets used for the management of neurotrauma (vagal nerve, periaqueductal grey, spinal cord), for which the modulation of autonomic function is a commonly proposed mechanism, is unclear [6,8,14,16,17,19,35]. However, given the established reciprocal connections between these targets, it is likely that diverse overlapping processes are elicited with each stimulation target [14,26,32]. 

This study also gives credence to prior clinical and laboratory studies that suggest a potential role for neuromodulatory therapies in the management of SCI and its sequelae [6,11,12,13,15]. Although the elucidation of precise mechanisms of action of NRM LFS was beyond the scope of this work, the implication of cell markers known to be involved in inflammation and repair is intriguing, since these processes have been commonly targeted before in attempts to restore function following SCI [13,25,36]. Neuromodulatory treatments of SCI, in general, would be welcome by the clinical community, since deep brain and spinal cord stimulation and the associated devices are already accepted as safe and effective for various other neurologic indications, reducing potential regulatory hurdles to their approval for SCI [10]. 

One limitation of this study was the analysis of a single time point at three days. This time point was used because it allowed us to obtain a broad snapshot of protective and restorative processes at a critical post-injury period when these processes are often at their peak [2,7]. Additionally, our prior studies found molecular, genetic, and behavioral effects of NRM LFS on injured spinal cord to be maximal when applied during this post-injury period. Another limitation was lack of behavioral data, which was not included due to the short post-injury survival period used here and because our prior work has shown that the therapeutic effect of NRM LFS reveals itself over longer experimental periods [6,16,17]. Another limitation is that, because this was a correlational study, the cellular effects, if any, that would contribute to or result from the improvements in functional recovery are unclear. 

## 5. Conclusions

A satisfactory treatment for pain and other signs of incomplete SCI does not yet exist but will ultimately require not only inhibiting detrimental secondary processes, such as edema and inflammation, but also recruiting cells that support regeneration. The targeting of brainstem centers that have multiple beneficial effects is a potential means towards accomplishing these goals, with the major strategic advantage that a single existing clinical tool is used. Prior to clinical application of this therapy, however, it will be important to better define the treatment’s mechanisms and their influence on recovery, for both the NRM and for various other promising stimulation targets. This approach can not only provide hope for improving the devastating outcomes associated with SCI but also improve our basic understanding of the brain’s central responses to SCI and their role in coordinating the necessarily incomplete process of natural recovery.

## Figures and Tables

**Figure 1 brainsci-09-00124-f001:**
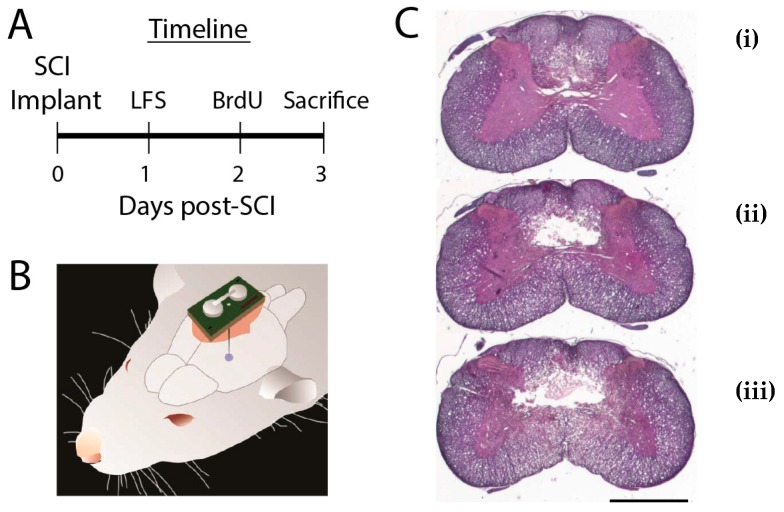
Experimental protocol. (**A**) Experimental timeline. Stimulators were implanted during the same procedure as the C5 contusion but activated one day later in the low-frequency electrical stimulation (LFS) group. (**B**) Schematic illustrating the size and position of the nucleus raphe magnus (NRM) stimulator relative to the rat brain. (**C**) Hematoxylin and eosin-stained sections showing a representative C5 lesion cavity 1 mm rostral to the lesion epicenter (**i**), at the lesion epicenter (**ii**), and 1 mm caudal to the lesion epicenter (**iii**). Scale bar is 1 mm. BrdU, bromodeoxyuridine; LFS, low-frequency stimulation; SCI, spinal cord injury.

**Figure 2 brainsci-09-00124-f002:**
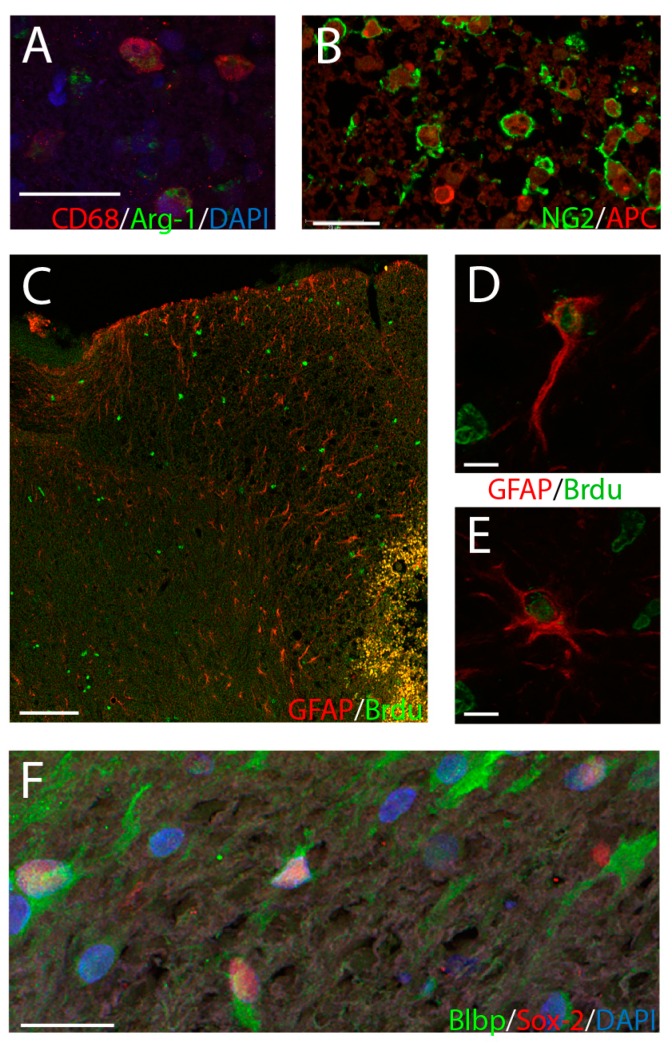
Immunohistochemical analyses. (**A**) Representative immunostain at 60× of an LFS-treated animal showing CD68-positive cells (red) co-labeled with arginase-1 (green) at the lesion. (**B**) Immunostain at 60× of an LFS-treated animal showing cells co-labeled for NG2 (green) and APC (red). (**C**) 10× micrograph through DWM and the lesion cavity showing GFAP (red) and BrdU (green) cells for a control animal. The lesion is seen in the bottom right corner of the image. (**D**) Unipolar GFAP/BrdU cell at 60× for an LFS-treated animal. (**E**) Multipolar GFAP/BrdU cell at 60× for an LFS-treated animal. (**F**) Immunostain at 60× for Blbp (green), Sox2 (red) and DAPI (blue) at the lesion for an LFS-treated animal. The specific immunostains examined and their corresponding colors are shown in the bottom right corner of each panel, except for D and E where this is shown in between the panels. Scale bars are 20 µm for A, B, and F, 100 µm for C, and 10 µm for D and E.

**Figure 3 brainsci-09-00124-f003:**
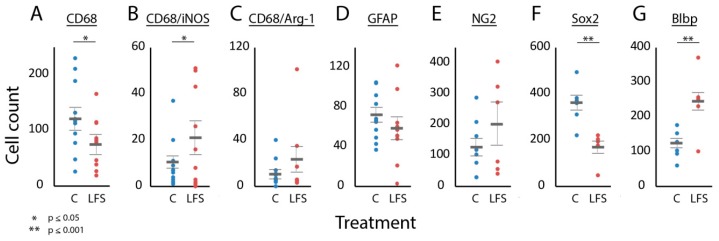
Effects of LFS of NRM on the expression of immune and neuroglial progenitor cell markers at the lesion. Scatter plots comparing cell counts for animals in the control (blue) and LFS (red) groups. Results are shown for CD68 (**A**), CD68/iNOS (**B**), CD68/Arg-1 (**C**), GFAP (**D**), NG2 (**E**), Sox2 (**F**), and Blbp (**G**). Only cell counts from DWM and GM are included. The thick horizontal gray bars show the mean of the counts. The thin horizontal gray bars show standard error. Asterixis above the graphs show statistical significance, if present, which was computed using bootstrap analyses. C, control; LFS, low-frequency stimulation.

**Figure 4 brainsci-09-00124-f004:**
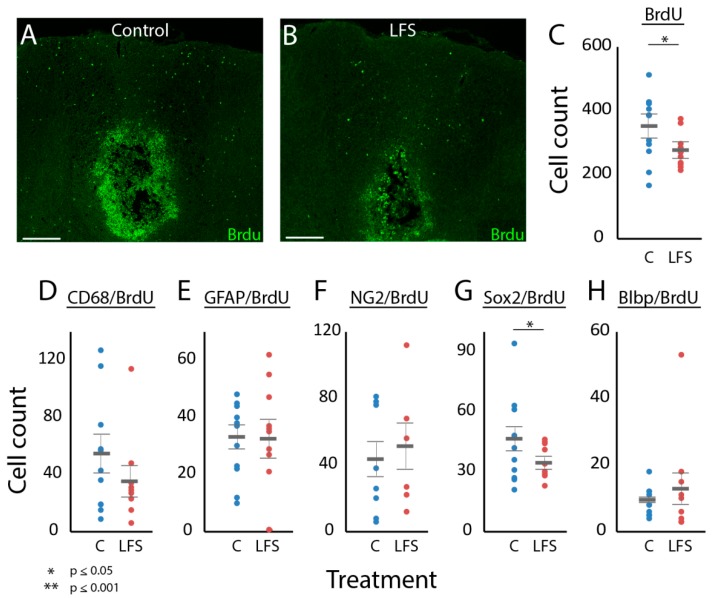
Effects of LFS of NRM on cellular proliferation at the lesion. Representative sections through DWM and GM of LFS-treated (**A**) and control (**B**) animals showing the expression of BrdU-positive cells surrounding the lesion cavity. The scatter plots compare cell counts of BrdU (**C**) and stains co-labeled with BrdU, namely CD68 (**D**), GFAP (**E**), NG2 (**F**), Sox2 (**G**), and Blbp (**H**), for animals in the control (blue) and LFS (red) groups. Only cell counts from DWM and GM are included. The thick horizontal gray bars show the mean of the counts and thin horizontal gray bars show standard error. Asterixis above the graphs denote statistical significance, which was tested for using bootstrap analyses. Scale bar for A and B is 200 µm. C, control; LFS, low-frequency stimulation.

**Table 1 brainsci-09-00124-t001:** Effects of LFS of NRM on cell counts for the different spinal cord locations examined.

Marker	Spinal Cord Location					
	**Rostral**		**Lesion**		**Caudal**	
	Control	LFS	Control	LFS	Control	LFS
BrdU	135 ± 14	147 ± 17	587 ± 50	519 ± 49	237 ± 24	209 ± 21
CD68	33 ± 8	34 ± 7	300 ± 39 *	224 ± 28 *	90 ± 21	74 ± 11
GFAP	175 ± 19	206 ± 31	187 ± 29	185 ± 29	114 ± 20	101 ± 23
	**DWM**		**GM**		**VWM**	
	Control	LFS	Control	LFS	Control	LFS
BrdU	105 ± 9	88 ± 11	239 ± 24 *	187 ± 16 *	243 ± 27	234 ± 22
CD68	61 ± 13 *	39 ± 8 *	114 ± 15 *	71 ± 8 *	125 ± 11	114 ± 12
CD68/iNOS	4 ± 2 *	11 ± 4 *	6 ± 2	10 ± 3	6 ± 2	9 ± 3
CD68/Arg-1	5 ± 1	9 ± 3	8 ± 3	13 ± 3	8 ± 2	14 ± 5
GFAP	28 ± 6	28 ± 7	73 ± 11	64 ± 10	86 ± 12	93 ± 12
NG2	81 ± 22 *	149 ± 39 *	46 ± 9	48 ± 10	54 ± 10	55 ± 15
Sox-2	101 ± 16	91 ± 13	155 ± 22 *	78 ± 12 *	128 ± 23 *	81 ± 17 *
Blbp	68 ± 15	102 ± 21	51 ± 8 *	140 ± 25 *	62 ± 15	107 ± 21

Shown are mean (± SE) counts for the various cellular markers examined. The top rows show the counts along the rostro-caudal axis of the spinal cord. The bottom rows show counts at the injury site for the three different spinal cord sub-regions analyzed. *P* values were estimated for individual comparisons using bootstrap analysis and comparisons with *p* ≤ 0.05 are marked with *. BrdU, bromodeoxyuridine; DWM, dorsal white matter; LFS, low-frequency stimulation; VWM, ventral white matter.

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
