# Peer review of "Cellular Changes in Injured Rat Spinal Cord Following Electrical Brainstem Stimulation"

_brainsci, 2019, doi:10.3390/brainsci9060124_

Reviewer 1 Report

The aim of this study was to evaluate the effect of low frequency stimulation (LFS) of the hindbrain’s nucleus raphe magnus (NRM) on the proliferation of immune and neuroglial progenitor cells in the subacute of spinal cord injury in rats with a partial contusion at the 5th cervical segment of the spinal cord. A group of rats received intermittent LFS 24 hours after SCI for the following 48 hours while the control group did not received any stimulation. Thus, the authors compared the two groups for the major populations of the immune and neural progenitors as well as BrdU (injected 24 before to sacrifice the animals). The authors concluded that LFS of NRM affects the subacute phase of SCI reducing inflammation and promoting neuronal maturation.  

I found the study well written and easy to follow with an appropriated introduction. However, the results section does not confirm what the authors stated in the abstract. The results section and the conclusion are not supported by the data presented here. Unfortunately, I found the description of the data particularly misleading since the data themselves are absolutely inconclusive  on the effect of the LFS on NRM. Thus, there are some major concerns.

Major concerns:

1) Lines from 169 to 194. Figure 3 does not match with the results section description:

- figure 3A and 3B described as ‘CD68-positive cells were significantly lower in LFS vs. control ‘ is clearly not;

-figure 3B described as ‘CD68-positive cells co-labeled with iNOS were higher in the LFS animals’ a it is clearly not;

- Figure 3C as CD68 cells co-labeled with arginase-1 trended higher (p = 0.20)’ that is not true.

-Figure 3F and 3G. Same as before. The description in the result section does not match with the graphs in the figure;

-Furthermore the word ‘trend’ in this study does not scientifically mean anything. Indeed, sometime the trend was in favor of the LFS and sometimes in favor of the controls with no effect depending directly from the stimulation;

-Lastly I did not really understand why the authors pooled together the data from the dorsal withe matter and the grey matter. They justified this choice by ‘Given these findings and the proximity of DWM and GM to the site of impact’ (line160). The data should have been analysis separately (figure 2) or all area pooled together. Only these analyses should then drive conclusion on cell proliferation;

2) Lines from 169 to 194. Again description of Figure 4 in the results does not match with the graphs in the figure;

- ‘The total BrdU count was lower at the lesion for rats treated with LFS of NRM compared to the

197 control animals (Figure 4A-C)’ (lines 196-197). The graph in 4C shows the contrary.

-authors claim that Figure 4D ‘CD68 and BrdU co-labeled cells trended lower, but did not achieve statistical significance (Figure 4D)’(lines 199-200). Again, the graph shows actually the opposite;

- ‘Univariate analysis indicated a significant relation between LFS and changes

 in the expression of cells co-localized with Sox2 and BrdU, which were lower in the LFS animals(Figure 4G) (lines 200-202)’. Sox2/BrdU cells are significantly more numerous in the LFS than in the control group;

-for all the rest of the data of this figure the word ‘trend’ is again used and like before it is misleading.

3)  The conclusions could not be supported by Figure number 3 and 4 since the descriptions of these figures in the results section are inappropriate. Even in case of wrong data representations (or data have been wrongly analyzed),  this study does not reveal any cellular mechanisms underlying a possible therapeutic use of LFS of NRM outcome after SCI. The discussion of the paper is based on assumptions that the immunoresponse after SCI is reduced by the LFS and simultaneously neuronal maturation is promoted. Further experiments are definitly needed to 

Minor concerns:

1)      Line 16. It should be clarified where the cellular proliferation will be evaluated;

2)      Line 34. The word ‘destruction’ should be replaced by a more appropriated word like damage or changes;

3)      Lines51-56: The paragraph is an overstatement of the authors that should be turned down . The raphe does not project to all areas of the brain and the spinal cord and there is not a theory of a raphe repair model, even just considering the spinal cord injury field. This brain area is subject to studies and its stimulation has effects on specific aspects of the impairments (of the sensory, motor and autonomic systems);

4)      Line 87. The word demonstrating must be replaced. Schematic illustrating may be more appropriate ;

5)      Line 87. The word ‘cranium’ should be replaced by the word ‘brain’;

6)      Figure 1C is never cited in the text;

7)      Figure 2. The indications of the markers in the panels will be better if are similar in font size;

8)      Line 197. ‘ substances’ should be replaced by a different word as markers

Author Response

The aim of this study was to evaluate the effect of low frequency stimulation (LFS) of the hindbrain’s nucleus raphe magnus (NRM) on the proliferation of immune and neuroglial progenitor cells in the subacute of spinal cord injury in rats with a partial contusion at the 5th cervical segment of the spinal cord. A group of rats received intermittent LFS 24 hours after SCI for the following 48 hours while the control group did not received any stimulation. Thus, the authors compared the two groups for the major populations of the immune and neural progenitors as well as BrdU (injected 24 before to sacrifice the animals). The authors concluded that LFS of NRM affects the subacute phase of SCI reducing inflammation and promoting neuronal maturation.  

I found the study well written and easy to follow with an appropriated introduction. However, the results section does not confirm what the authors stated in the abstract. The results section and the conclusion are not supported by the data presented here. Unfortunately, I found the description of the data particularly misleading since the data themselves are absolutely inconclusive  on the effect of the LFS on NRM. Thus, there are some major concerns.

 Major concerns:

 1) Lines from 169 to 194. Figure 3 does not match with the results section description:

- figure 3A and 3B described as ‘CD68-positive cells were significantly lower in LFS vs. control‘ is clearly not;

-figure 3B described as ‘CD68-positive cells co-labeled with iNOS were higher in the LFS animals’ a it is clearly not;

- Figure 3C as CD68 cells co-labeled with arginase-1 trended higher (p = 0.20)’ that is not true.

-Figure 3F and 3G. Same as before. The description in the result section does not match with the graphs in the figure;

-Furthermore the word ‘trend’ in this study does not scientifically mean anything. Indeed, sometime the trend was in favor of the LFS and sometimes in favor of the controls with no effect depending directly from the stimulation;

We truly apologize but the data was entirely misrepresented in every scatter plot in Figures 3 and 4, the most important figures of the paper. The x-axis labels of the scatter plots were reversed in error, i.e. ‘C’ and ‘LFS’ were reversed in every case. We have corrected this in the new figures and confirmed that the figures are now representative of the data. We have also ensured that there are no other errors present in the figures. We thank the Reviewer for pointing out these important mistakes. References to a “trend” are now entirely omitted, and differences are appropriately stated to be not statistically significant.

 -Lastly I did not really understand why the authors pooled together the data from the dorsal withe matter and the grey matter. They justified this choice by ‘Given these findings and the proximity of DWM and GM to the site of impact’ (line160). The data should have been analysis separately (figure 2) or all area pooled together. Only these analyses should then drive conclusion on cell proliferation;

We believe that our motivation for pooling the data was not explained well. We did this not only because cellular effects of LFS were mainly observed in DWM and GM, but also because lesion cavities were confined to these two areas on histology. We clarified this further in sections 3.1 and 3.2 of the manuscript. 

The observation that effects of LFS are in close proximity to the injury site is something we have observed before using different experimental paradigms, which is why we feel justified in pooling the data here. Nevertheless, for the sake of completeness, we did include analysis with all spinal cord subregions included in the Supplement (In text citation on line 169). As can be seen, in general, our results hold, though the statistical effects are not as strong, likely from the incorporation of VWM tissue were effects of LFS were lesser than for the other two subregions. 

 2) Lines from 169 to 194. Again description of Figure 4 in the results does not match with the graphs in the figure;

- ‘The total BrdU count was lower at the lesion for rats treated with LFS of NRM compared to the control animals (Figure 4A-C)’ (lines 196-197). The graph in 4C shows the contrary.

- authors claim that Figure 4D ‘CD68 and BrdU co-labeled cells trended lower, but did not achieve statistical significance (Figure 4D)’(lines 199-200). Again, the graph shows actually the opposite;

- ‘Univariate analysis indicated a significant relation between LFS and changes in the expression of cells co-localized with Sox2 and BrdU, which were lower in the LFS animals (Figure 4G) (lines 200-202)’. Sox2/BrdU cells are significantly more numerous in the LFS than in the control group;

-for all the rest of the data of this figure the word ‘trend’ is again used and like before it is misleading.

Once again, we apologize for our mistake. Figure 4, as well as its description in the Results, are now representative of the data. All phrases with the word “trend” have been corrected or omitted.

 3)  The conclusions could not be supported by Figure number 3 and 4 since the descriptions of these figures in the results section are inappropriate. Even in case of wrong data representations (or data have been wrongly analyzed), this study does not reveal any cellular mechanisms underlying a possible therapeutic use of LFS of NRM outcome after SCI. The discussion of the paper is based on assumptions that the immunoresponse after SCI is reduced by the LFS and simultaneously neuronal maturation is promoted. Further experiments are definitly needed. 

In addition to correcting the mistakes in Figures 3 and 4, we admit that we were a little aggressive in our final interpretations of the data. To address this, we have changed the conclusions from “our work shows that LFS of NRM in subacute SCI may modify inflammatory cell types and promote neuronal maturation” to “our work shows that LFS of NRM in subacute SCI influences the proliferation of cell types implicated in inflammation and repair”

 Minor concerns:

1)      Line 16. It should be clarified where the cellular proliferation will be evaluated;

This was added to Line 21. 

2)      Line 34. The word ‘destruction’ should be replaced by a more appropriated word like damage or changes;

Changed

3)      Lines51-56: The paragraph is an overstatement of the authors that should be turned down. The raphe does not project to all areas of the brain and the spinal cord and there is not a theory of a raphe repair model, even just considering the spinal cord injury field. This brain area is subject to studies and its stimulation has effects on specific aspects of the impairments (of the sensory, motor and autonomic systems);

We agree that the NRM does not project throughout the CNS, but serotonergic axon terminals are essentially ubiquitous in the CNS and all arise from one or other of the raphe nuclei in the midbrain or hindbrain. We have tried to make the distinction between the NRM and the raphe system as a whole clearer in the text. Several of the other statements have been turned down. Instead of stating that LFS improves those SCI sequelae, we use the word “influences”. “All parts of brain and spinal cord” changed to “fibers that diffusely project to most areas of the nervous system”, and “the NRM provides a major spinal projection for this system”. Additionally, instead of stating “raphe repair model”, we now state “proposed repair model”. We left our mention of our TBI study in just because we feel like it strengthens the general argument for brainstem areas involved in repair. 

4)      Line 87. The word demonstrating must be replaced. Schematic illustrating may be more appropriate;

Changed

5)      Line 87. The word ‘cranium’ should be replaced by the word ‘brain’;

Changed

6)      Figure 1C is never cited in the text;

Line 107

7)      Figure 2. The indications of the markers in the panels will be better if are similar in font size;

We checked and it appears that the indications are the same size and font. 

8)      Line 197. ‘substances’ should be replaced by a different word as markers

Changed

Reviewer 2 Report

The authors present a very interesting study of the effect of low frequency stimulation of the raphe nuclei on cellular changes after SCI. This builds on the authors previous work and provides some important information regarding the mechanic of this approach and suggests a role in modulation inflammation, an important component  of recovery after SCI. While it is correlational and has limitations as the authors point out it provides some novel insights.

However, there is one major issue with this manuscript and that is to do with the presentation of the data. in the text with respect to figure 3 what  is stated in the text regarding these results appears to be opposite to what is presented in the figure i.e CD68-positive cells were significantly lower in LFS vs. control animals (Figure 3A) but in the figure the average appears greater in the LFS treated cells. This appears to occur for other cell analyses as well. Same for Figure 4, the data presented in the figure appears to be opposite to how it is described in the text. Has the data in the graph been presented incorrectly. This needs to be clarified.

Regarding the image analysis, it is not clear how many areas have been counted for each animal. Is it only one 60X image per area per animal. This needs to be clarified.

Another thing that needs clarification is where the authors refer to a partial injury. What do the authors  mean by this. To this reviewer a partial injury is a hemisection or unilateral contusion. Do they mean that the animals still have some function or naturally regain some function as would be expected for a contusion injury of this type?  Some info about the severity of this type of injury in their hands, in terms of function or other parameters should be given.

Author Response

The authors present a very interesting study of the effect of low frequency stimulation of the raphe nuclei on cellular changes after SCI. This builds on the authors previous work and provides some important information regarding the mechanic of this approach and suggests a role in modulation inflammation, an important component of recovery after SCI. While it is correlational and has limitations as the authors point out it provides some novel insights.

 However, there is one major issue with this manuscript and that is to do with the presentation of the data. in the text with respect to figure 3 what is stated in the text regarding these results appears to be opposite to what is presented in the figure i.e CD68-positive cells were significantly lower in LFS vs. control animals (Figure 3A) but in the figure the average appears greater in the LFS treated cells. This appears to occur for other cell analyses as well. Same for Figure 4, the data presented in the figure appears to be opposite to how it is described in the text. Has the data in the graph been presented incorrectly. This needs to be clarified.

We truly apologize but the data was entirely misrepresented in every scatter plot in Figures 3 and 4, the most important figures of the paper. The x-axis labels of the scatter plots were reversed in error, i.e. ‘C’ and ‘LFS’ were reversed in every case. We have corrected this in the new figures and confirmed that the figures are now representative of the data. We have also ensured that there are no other errors present in the figures. We thank the Reviewer for pointing out these important mistakes. 

 Regarding the image analysis, it is not clear how many areas have been counted for each animal. Is it only one 60X image per area per animal. This needs to be clarified.

This is valuable information that was omitted from our original version. To clarify this, we added this to the current version: “For each of the three sections, a sampling grid of 225 X 225 mm was placed over each of the three contoured subregions. Within each grid box an area of 75 X 75 mm was selected by the software as the counting frame and cell markers were counted at 60X. The number of counting frames used for each subregion varied between sections, but typically ranged from 60 to 90 for DWM, 120 to 180 for GM, and 190 to 270 for VWM.”

 Another thing that needs clarification is where the authors refer to a partial injury. What do the authors mean by this. To this reviewer a partial injury is a hemisection or unilateral contusion. Do they mean that the animals still have some function or naturally regain some function as would be expected for a contusion injury of this type?  Some info about the severity of this type of injury in their hands, in terms of function or other parameters should be given.

Some of this information was included in the initial version (lines 149-150): Following moderate C5 contusion, flaccid, incomplete quadriplegia was noted in all animals, which showed some recovery of function by the time the animals were euthanatized. We have further expanded on this by stating our observation that lesion cavities on histology were confined to DWM and GM. Please tell us if further clarification of the partial injury is required. We have also substituted “incomplete” or “bilateral incomplete” for “partial”, as appropriate, to indicate that the lesion is not a hemisection or a unilateral contusion. The Methods states that the lesions was made in the midline.

Round  2

Reviewer 1 Report

Clearly the mistakes on figures 3 and 4 determined the misinterpretation of the study. Now that this has been corrected, the results looks consistent and the conclusions are supported by the results. This reviewer is now satisfied by the changes the authors have made on the draft. 

Reviewer 2 Report

The authors have addressed my concerns and I see no other issues that need addressing. I also feel the changes made based on the other reviewers comments have also improved the manuscript.